# Biodiesel Production from Waste Oil Catalysed by Metal-Organic Framework (MOF-5): Insights on Activity and Mechanism

Francesco Taddeo [1], Rosa Vitiello [1], Vincenzo Russo [1], Riccardo Tesser [1], Rosa Turco [1,2,*] and Martino Di Serio [1]

1    Department of Chemical Sciences, Monte Sant'Angelo Campus, University of Naples "Federico II", Via Cinthia 4, 80126 Naples, Italy
2    Institute for Polymers, Composites and Biomaterials, National Council of Research, Via Campi Flegrei 34, 80078 Pozzuoli, Italy
*    Correspondence: rosa.turco@unina.it; Tel.: +39-081674011

**Abstract:** The activity of MOF-5-based solids has been exploited in the simultaneous transesterification and esterification of acid vegetable oils. For this purpose, three different types of MOF-5 have been synthesized and characterized, and then tested in the above-mentioned reactions. It has been demonstrated that the "regular MOF-5" was a suitable catalyst for biodiesel synthesis from waste oil also, rich in FFA (Free Fatty Acids). Moreover, to identify the true structure that acts in the reactions and possible structural modifications due to the presence of alcohols, proper studies have been performed. The results have evidenced a distortion of the regular structure of MOF-5 due to the breakage of some zinc bonds between the cluster and organic framework.

**Keywords:** metal organic frameworks; MOF-5 activity; transesterification reaction; waste oils





## 1. Introduction

According to the International Energy Agency report of 2016 on "Renewable Energy Term Market", the renewable energy sources hold a noteworthy position in the global market of energy consumptions [1]. This increment can be justified considering the depletion of fossil resources and environmental concerns connected to their use. In this scenario, biodiesel represents a promising alternative to face these challenges. Biodiesel, also known as Fatty Acid Methyl Ester (FAME), corresponds to a mixture of methyl esters, derived by the transesterification reaction of vegetable and animal oils or fats with methanol (Figure 1a).

Many advantages are commended by the use of biodiesel in comparison with fossil oil as the energy source: biodegradability, renewability, higher flash point and lubricity, absence of sulphur, reduced emissions, and pollutants [2].

However, there are some limitations for the use of biodiesel compared with petro-diesel; for example, the feedstock costs have been estimated to share about 75–88% of total production costs and the necessity to substitute the classical vegetable oil used for its production with a raw material that is not in competition with food.

Some strategies have been investigated and proposed to overcome the barrier of costs and get competitive the use of biodiesel as an alternative to diesel from the economical point of view: the improvement of the production technologies, the development of better catalysts, and the use of alternative feedstocks [3]. Several raw materials can be considered to produce biodiesel, such as animal fats, vegetable oils, and algae oil. The choice depends on the country and the availability of local resources; for example, soybean oil prevails in the USA, whereas palm and rapeseed prevail in Europe and tropical regions. However, for edible oils, the food-energy debate has been raised for developing countries. Therefore,

non-edible oils represent a sustainable answer to this issue. In this regard, the transport oil costs, connected to the use of foreign non-edible oil, must be considered. The logic lies in the application of cheaper local feedstock. Waste oils, derived from restaurants and household cooking, are an economical and local source of oil for biodiesel synthesis, which can allow the reduction of production costs and resolve the disposal problem of the waste. However, the waste oils or fats and non-edible crops oil have higher FFA (Free Fatty Acid) and water content; these can injure the yield and quality of the produced biodiesel due the occurrence of side reactions (Figure 1b).

**(a)**

**(b)**

**Figure 1.** (**a**) Transesterification reaction of triglycerides with methanol. (**b**) FFA (Free Fatty Acids) esterification reaction.

This setback is mainly noticeable for most economic process based on alkali-catalyzed transesterification with methanol. The presence of FFA and water promotes the soap formation by methoxide specie instead of transesterification reaction with the triglyceride molecule. Therefore, in these cases, the opportune pre-treatment step, based on the esterification reaction, is needed [4].

Moreover, Brønsted acids can be used as catalysts for the transesterification, even if the reaction is slower in this case in addition to corrosivity problems due to the presence of strong acids [5]. The transesterification with supercritical methanol is another advantageous answer. For this technology, catalysts are not employed; thus, the pre-treatment is not needed. However, a very large content of alcohols and severe values of temperature and pressure are required [6].

The use of catalysts capable to promote simultaneously both the transesterification of triglycerides and esterification of FFA in methyl esters (FAME) would simplify the whole process, achieving a reduction of both investment and process costs. Homogenous and heterogenous catalysts have been proposed in the literature as able to promote both reactions [7]. The use of heterogeneous catalysts represents the advantageous choice to get a more sustainable process through the recovery of the catalyst at the end of the reaction cycle, in obedience to some Green Chemistry Principles [8].

Various examples of bifunctional catalysts of esterification and transesterification reactions can be found in the literature. SrO-ZnO/Al$_2$O$_3$ has been involved in biodiesel production and in the esterification of oleic acid [9], as well as ZnO nanoparticles used in the transesterification of $\alpha$-ketocarboxylic esters [10]. Catalysts based on MgO and hydrotalcites have also been used in the transesterification reaction of soybean oil with methanol by Di Serio et al. [11]. The reaction for the production of biodiesel has also been catalyzed by calcium oxide being a promising catalyst [12], and, in addition to these, MOFs are also a class of catalysts suitable for esterification and transesterification reactions for

the production of biodiesel, as reported by Jamil et al. [13]. Table 1 shows a comparison between different catalysts used in the production of biodiesel.

**Table 1.** Catalysts for biodiesel production.

| Catalyst | Substrate | Catalyst Loading (wt%) | MeOH/ Substrate (mol/mol) | $Y_{FAME}$ (%) | Reference |
|---|---|---|---|---|---|
| Vanadyl phosphate (VOP) | Soybean oil | 6.5 | 27/1 | 80.0 | Di Serio et al. (2007) [14] |
| MOF-5 | Glyceril triacetate | 16.5 | 180/1 | 9.2 | Chen et al. (2014) [15] |
| MOF-5-ED | Glyceril triacetate | 3.0 | 30/1 | 90.0 | Chen et al. (2014) [15] |
| IRMOF-5-ED | Glyceril triacetate | 16.5 | 180/1 | 99.9 | Chen et al. (2014) [15] |
| MOF-5-ED | Glyceril butyrate | 10.0 | 30/1 | 99.0 | Chen et al. (2014) [15] |
| IRMOF-5-ED | Glyceril butyrate | 10.0 | 30/1 | 99.0 | Chen et al. (2014) [15] |
| IRMOF-5-ED | Glyceril triacetate | 5.5 | 30/1 | 57.6 | Chen et al. (2014) [15] |
| Hβ-zeolite | Pongamia pinnata oil | 11.5 | 10/1 | 59.0 | Karmee and Chadha (2005) [16] |
| Montmorillonite K-10 | Pongamia pinnata oil | 11.5 | 10/1 | 47.0 | Karmee and Chadha (2005) [16] |
| ZnO | Pongamia pinnata oil | 11.5 | 10/1 | 83.0 | Karmee and Chadha (2005) [16] |
| UiO-66 | Soybean oil | 11.0 | 40/1 | 98.5 | Zhou et al. (2016) [17] |
| MOF-5 | Waste cooking oil | 0.75 | 36/1 | 82.0 | Ben-Youssef et al. (2021) [18] |

Metal-Organic Frameworks (MOFs) have recently been attracting growing attention as heterogenous catalysts due to their notable versatility [19,20]. They consist of micro-mesoporous hybrid materials, composed by metal or metal-cluster nodes bridged by organic linkers in two- or three-dimensional frameworks [21]. These structures, also named porous coordination polymers (PCPs), correspond to a metal–ligand coordination with an organic ligand containing voids or channels. The interest in MOFs from academic and industrial research can be justified considering the concomitant presence of three key features: high crystallinity, porosity, and strong metal–ligand interactions. The microporous structure, due to pores with a specific volume of about $2 \text{ m}^2 \text{ g}^{-1}$, provides very high specific surfaces up to $6000 \text{ m}^2 \text{ g}^{-1}$, allowing their application for gas separation and storage, explosive vapors, and nerve agent pre-concentrator [22]. In particular, in the literature, a value of $3800 \text{ m}^2 \text{ g}^{-1}$ is reported for the MOF-5 area and $1.55 \text{ m}^2 \text{ g}^{-1}$ for pore volume [23].

An important property of MOFs is their modularity: the pore dimension, shape, and chemical environment can be tailored by the choice of a couple of building blocks (metal and organic linker), in addition to the solvent, and how they are interconnected [24–26].

Many metals can be suitable as nodes in MOFs, such as transition, alkaline and alkaline-earth metals, whereas polycaboxylic aromatic molecules, bipyridines, and polyaza-heterocycles such as imidazole represent the most used organic linkers.

Several syntheses of MOFs with different structures were described in the literature; in particular, IRMOF [27], ZIF-8 [28,29] and MIL series [30] are the better-known examples.

In the field of the catalysis, they have been recommended for being active for many reactions, such as condensation of carbonyl groups, heterocycles synthesis, oxidations, fine chemicals synthesis, and photocatalysis [31].

Chizallet et al. [28] reported, for the first time, the use of a MOF (ZIF-8) for the transesterification reaction of vegetable oil. ZIF-8 showed significant reactivity in rapeseed oil transesterification with many linear alcohols. Cirujano et al. [32] showed the activity of Zr-containing UiOs-based-MOFs (UiO-66 and UiO-66-NH₂) for the esterification reaction of several fatty acids with methanol. However, a decline in yields was reported to increase the complexity of the fatty acid chain in terms of length, ramification, and unsaturation.

The same conclusion was shown by Liu et al. [33] for biodiesel production from fatty acids with methanol in the presence of NENU-3a.

Chen et al. [15] reported the application of amine-functionalized MOF (MOF-5, IRMOF-10, and MIL-53(Al)-NH$_2$) as a solid base catalyst for the transesterification of triacetin and glyceryl tributyrate with methanol, showing an excellent performance with about 99% of conversion at low temperature (50 °C) and time (3–4 h).

Starting from this literature analysis, this work aimed at the study of MOF-based materials as active catalysts for biodiesel synthesis from waste oils. For this purpose, the chosen catalyst could be able to promote both reactions of FFA esterification and triglyceride transesterification with methanol. Therefore, in this work, a MOF-5 has been properly synthesized and characterized to confirm the structure by X-ray diffractometry, UV-VIS spectroscopy, and thermogravimetric analysis, and it has been tested in simultaneously acid oil esterification transesterification reactions for the biodiesel production to evaluate the activity.

The use of MOF-5 has already been proposed in the literature to promote the transesterification and esterification reactions, though never simultaneously. It has also never been discussed how the active sites acted in the reaction. Therefore, in this work, the type of mechanism and the possible active sites were studied to clarify the behavior of this solid as a heterogeneous catalyst. It is well known that two crystalline structures for MOF-5 can be possible, such as cubic and tetragonal, depending on the location of the zinc, which can exist separately or coexist together in a mixed structure (blend). Therefore, both structures have been prepared in this work for a general comparison of activity for the MOF-5-based solids in the studied reactions.

Subsequently, to identify if the catalyst undergoes structural changes when it is used in the reaction, a proper study has been performed, keeping the solid in contact with methanol at different temperatures and investigating the final structure, in addition to the activity. Based on the collected data, a conceivable mechanism has been hypothesized.

Finally, stability MOF has been verified in the reaction with proper tests of leaching and recycling.

## 2. Results and Discussion

### 2.1. Synthesis of the Catalyst

In this work, three solids based on MOF-5 with different structures were synthesized, as reported in Table 2, together with the obtained final yields. MOF-5 with cubic, tetragonal, and a mixture of them were prepared. From Table 2, it is evident that high yields have been reached, confirming the validity of the synthesis methods applied. Moreover, the cubic and tetragonal structure-based MOF-5s were synthesized with the aim of a comparison in the activity in the simultaneous reaction of the transesterification and esterification of acid oil with the blend MOF-5.

**Table 2.** Summary of experimental conditions for MOF-5-based solids investigated. H$_2$BDC = 2-hydroxy-1,4-benzenedicarboxylic acid; TEA = triethylamine; DMF = *N*,*N*-dimethylformamide.

| Sample | Structure | Zn(NO$_3$)$_2$ (g) | H$_2$BDC (g) | H$_2$O$_2$ (mL) | TEA (mL) | DMF (mL) | T (°C) | Time (h) | T$_{activation}$ (°C) | Yield (%) |
|---|---|---|---|---|---|---|---|---|---|---|
| B-MOF-5 | Blend | 4.72 | 1.01 | / | / | 147 | 130 | 4 | 60 | 74 |
| T-MOF-5 | Tetragonal | 1.19 | 0.34 | 2 | 2.5 | 40 | R.T. | 1 | 180 | 91 |
| C-MOF-5 | Cubic | 3.87 | 2.11 | / | 2.6 | 104 | 100 | 7 | 115 | 89 |

### 2.2. Characterisation of the Catalyst

Two different crystal structures are possible for a MOF-5-based solid. The first one is a cubic structure, where the inorganic clusters of [Zn$_4$O]$^{6+}$ are joined to an octahedral array of benzene-1,4-dicarboxylate (BCD) groups, forming a cubic Zn$_4$O(BCD) framework with 12.94 Å as the distance between the centers of adjacent clusters. This structure corresponds to a form of MOF-5, known as "Regular". The second structure, which is common for

MOF-5 solids, is a tetragonal one, due to the presence of ZnO in the BCD framework, which causes a variation of axes length, forming a tetragonal structure.

Herein, the current structure for the synthesized MOF-5 was determined by XRD analysis. According to the literature, the principal difference in XRD profiles between tetragonal and cubic MOF-5 consists in the intensity of the reflections: (200) ($2\theta \approx 7.0°$) and (220) ($2\theta \approx 9.8°$). The higher intensity of the (200) reflection suggests the presence of a cubic structure, or a tetragonal one. The XRD pattern, shown in Figure 2a,b, revealed the presence of a tetragonal structure for blend MOF-5 (B-MOF-5), and confirms the tetragonal structure for the T-MOF-5 solid.

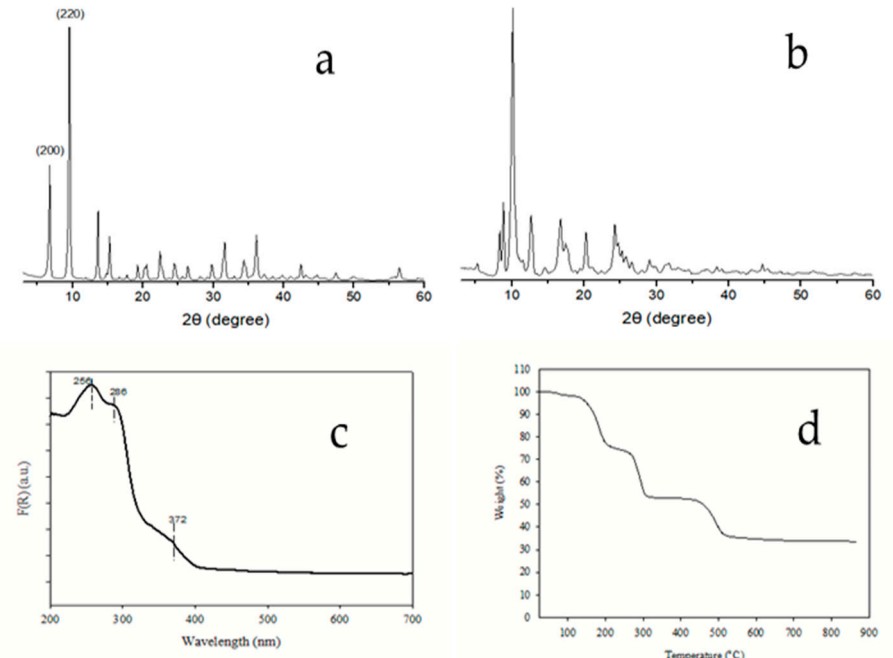

**Figure 2.** Characterization results for synthesized MOF-5-based solids: XRD patterns for blend (**a**) and tetragonal (**b**) MOF-5.; UV-DRS spectrum (**c**) and thermogravimetric profile (**d**) for MOF-5-B.

To be more precise, according to the literature, the principal difference in XRD profiles between tetragonal and cubic MOF-5 consists in the intensity of the reflections: (200) ($2\theta \approx 7.0°$) and (220) ($2\theta \approx 9.8°$). The higher intensity of the (200) reflection suggests the presence of a cubic structure, or a tetragonal one. From Figure 2a,b, it is evident that the intensity of the reflection at 9.8° is strikingly greater than the one at 7.0° for both B-MOF-5 and T-MOF-5. For the C-MOF-5 that has a cubic structure, there are uncertainties in the XRD spectrum, since, as discussed in the literature, this structure is very unstable and tends to change even when in contact with humidity.

Moreover, the predominance of the tetragonal structure for B-MOF-5 was confirmed by DRS analysis, as depicted in Figure 2a,b. Three main signals can be seen in the UV-vis spectrum: two peaks (256 and 286 nm) are usually associated to the electron's transitions $\pi \rightarrow \pi^*$ of the organic aromatic C6 ring. The third signal at 372 nm is distinctive of the type of the structure because it is usually assigned to the electronic transitions of ZnO and to $O^{2-}Zn^{2+} \rightarrow O-Zn^+$ ligand-to-metal transfer transition [34,35]. This signal confirmed the presence of ZnO in B-MOF-5, and it is detectable only for the tetragonal MOF-5.

In Figure 2c,d, the thermal decomposition of B-MOF-5 is reported too. The TGA profile shows four weight losses at different temperature values: the first one at 100 °C (2% wt.) can be associated to the evaporation of adsorbed moisture; the second (19% wt.), in the range of 100–240 °C, can be attributed to *N,N*-Diethyl formamide (DEF) removal; the remaining weight losses at higher temperatures reflect the collapse of the framework, which begins at about 240 °C. This lower thermal stability, observed for the synthesized MOF-5,

can be ascribed to the use of DEF as a solvent instead of *N,N*-Dimethyl formamide (DMF). According to Chen et al. [36], the use of DEF brings forward the structure decomposition at 250 °C rather than at 375 °C with DMF. Therefore, the first weight loss (18 wt%), observed in the range 2400–300 °C, can be assigned to the starting point of the decomposition of MOF-5, whereas the second one, around 400 °C, can be assumed due to the total collapse of the frameworks [37,38].

Scanning electron microscopy (SEM) was used to investigate MOF-5 morphology, shape, and surface structure. In Figure 3a, MOF-5 before the contact with methanol is reported. The images clearly show the presence of distorted cubic structures, typical of MOF-5 [39]. The catalyst after the contact with methanol is indicated with the letter B. In this case, the expected morphology of the MOF-5 that is lost and the break of the bonds are clearly visible.

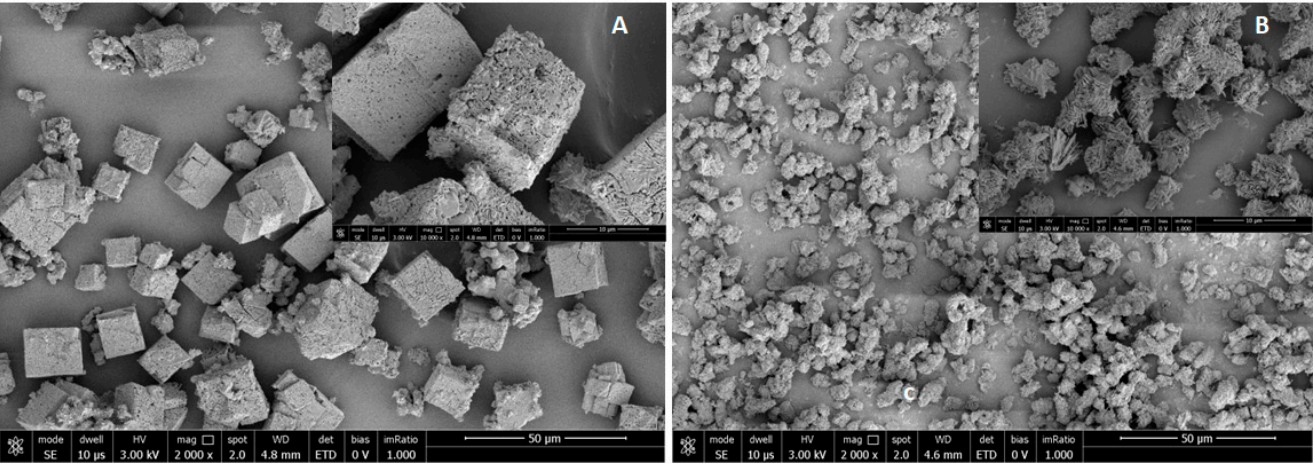

**Figure 3.** (**A**) MOF-5 before the contact with methanol. (**B**) MOF-5 after the contact with methanol.

*2.3. Catalytic Activity*

Figure 4 shows the results, obtained in the transesterification and esterification reactions of neutral/acid oil with methanol, in the presence of the different types of MOF-5-based solids. At first, blank experiments carried out with bare reagents (acid oil/methanol) without a catalyst showed that there is not any significative contribution to the reaction from the metal walls of the reactor. Regarding the transesterification/esterification reactions, the results highlighted that the structure of MOF-5 strongly influences both the transesterification and the esterification reactions. The better performance has been depicted for B-MOF-5 and T-MOF-5; the worst for C-MOF-5.

These results can be ascribed to the presence of channels with different dimensions, depending on the type of structure or lattice of the MOF. The distortion of the lattice, which is evident in the tetragonal ones (B-MOF-5 and T-MOF-5), creates a more open structure, facilitating the passage of molecules for the reaction, whereas for a cubic structure, the presence of a more closed reticle is assumed, which could create diffusion problems for more bulky molecules such as triglycerides.

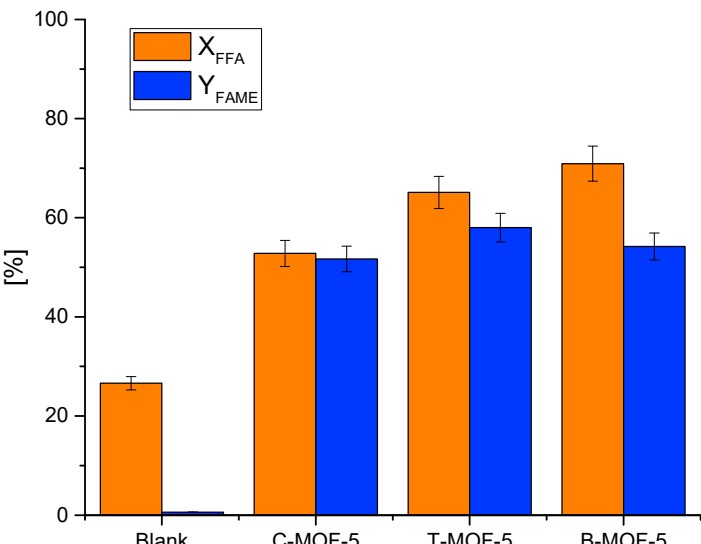

**Figure 4.** Results of FFA conversion, $X_{FFA}$, (%) and yields in methylesters, $Y_{FAME}$, (%) obtained with MOF-5 catalyst. Experimental conditions: reactants molar ratio = 3, temperature = 150 °C, catalyst amount = 5 wt%, reaction time = 3 h, (initial acidity number = 16.5 mgKOH/gsample or FFA = 10 wt%).

On the other hand, this is also evident from the fact that the results obtained with the tetragonal structure (T-MOF-5) are comparable with the blend (B-MOF-5), where this structure predominates (see Figure 4). The distortion of the lattice, which is evident in the tetragonal one, creates a more open structure, facilitating the passage of molecules for the reaction. However, these results appear to be slightly lower than those which have been reported in the literature for esterification/transesterification reactions in the presence of similar MOF-based catalysts. The difference may be ascribed to a larger complexity of the organic substrate investigated: neutral and acid vegetable oil in this work, and carboxylic acids and only glyceril tibutyrate in the literature. Both Cirujano [32] and Liu [33] separately stated a decrease in yield (%) for both reactions, increasing the length and unsaturation degree of fatty acids. Moreover, the adopted experimental conditions and the MOF-5 prepared in this paper have enabled a reduction in the amount of the catalyst required to reach a high value of yield. Nevertheless, the results shown here seem very promising, since they show that MOF-5 is an active catalyst for both the transesterification and esterification reactions and show it as a suitable catalyst for biodiesel production from waste oils. This finding is a novelty for the existing literature concerning the application MOF-based materials in the transesterification of acid oils.

In order to verify the heterogenous nature of the solid catalyst used for this work, the leaching effect was evaluated by a proper test using B-MOF-5, as it was described in the experimental section. For these tests, a slight increase in FAME yield was observed (25%) in respect to blank test yield (less than 5%); these results indicated a low leaching of the active phase in the reaction media.

Considering the instability of some MOF-5 structures with humidity, a study was conducted to identify which of the two structures was active in the transesterification reaction. These tests were carried out by placing them in contact with methanol at different temperatures, and the structure of the dried solid was studied through X-ray analysis and UV spectroscopy. The XRD pattern of the recovered solid (B-MOF-5) is reported in Figure 5a, which highlights that the contact with methanol promotes a lowering of the intensity of the reflection (200) at 2θ ≈ 7°, a shift of the reflex (220) from 2θ ≈ 9.8° to 2θ ≈ 8.7°, and the appearance of a new signal at 2θ ≈ 8.1°.

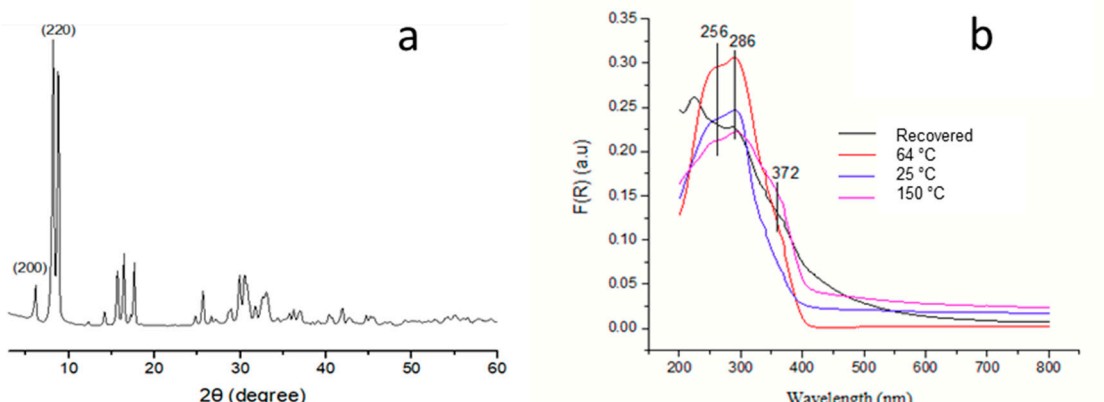

**Figure 5.** (**a**) XRD diffractogram of recovered solid (B-MOF-5), (**b**) DR-UV del Blend MOF-5 at different temperature and recovered.

This finding can be ascribed to a distortion of the regular structure of MOF-5 caused by the breaking of some Zn–O bonds between the cluster and the organic lattice (Figure 6), leading to a more open structure that proves to be still active, which is also evident from a comparison with the literature [40].

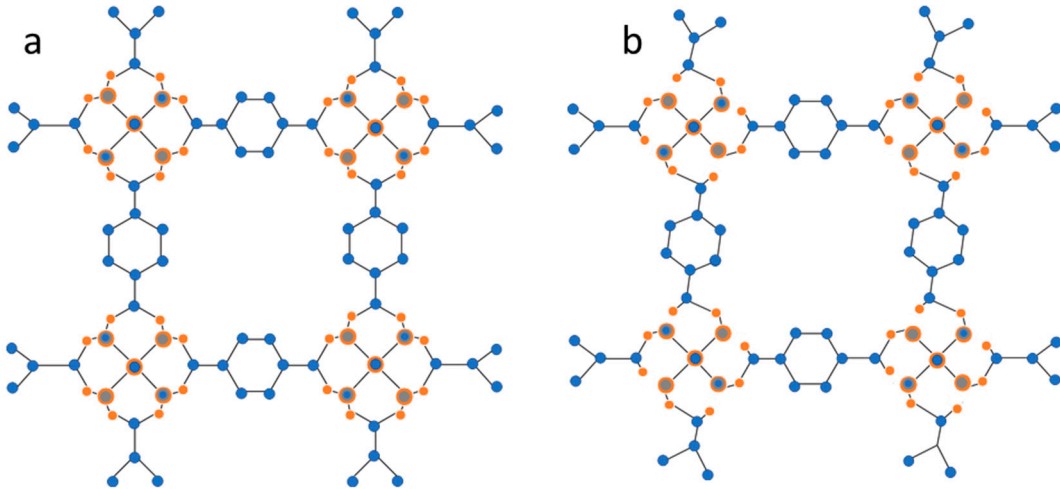

**Figure 6.** (**a**) Cubic structure of MOF-5 and (**b**) distorted structure of MOF-5 after contact with methanol. The atoms are represented in the following way: zinc in grey, oxygen in orange, carbon in blue.

In Figure 5, the DRS spectra related to B-MOF-5 both recovered at the end of the reactions (a) and after the contact with the bare methanol at different temperature values, 60 °C (b), 25 °C (c), and 150 °C (d). For all spectra, the characteristic signals of MOF-5 have been observed, as well as a broad shoulder at 372 nm, correlated with the tetragonal structure and also the electronic transitions of ZnO. This finding is especially prevalent for the solid treated with methanol. Therefore, in the light of the collected results, the following hypothesis has been formulated: the methanol in this reaction network acts to promote a modification of the structure of MOF-5 with the break of some bonds in the lattice, leading to a more open structure. This structure is stable, as has been confirmed by the reuse test and leaching tests and maintains its activity in the studied reaction for biodiesel production. The catalyst was reused for three cycles in a Parr Instrument reactor, and a decrease in the catalyst performance was not observed, obtaining almost the same results for the FAME yields in each test (see Figure S1 in Supplementary Materials). This hypothesis has been further confirmed by the results in Figure 7, where the activity in

transesterification and esterification reactions after the methanol contact are shown. The obtained results are comparable with that of fresh catalysts (see Figure 4).

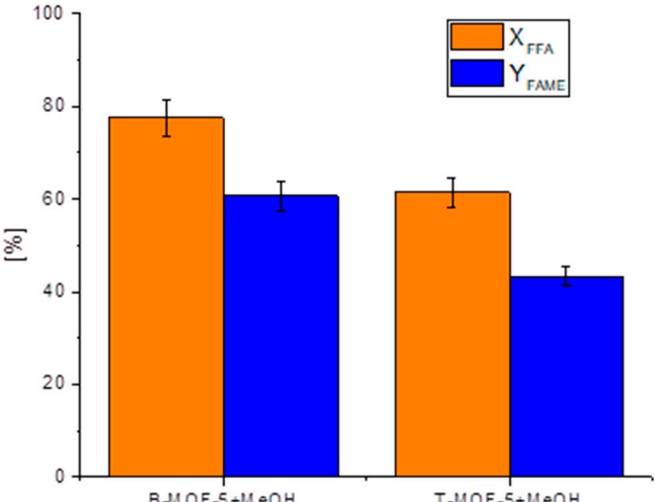

**Figure 7.** Results of FFA conversion, $X_{FFA}$, (%) yields in methyl esters, $Y_{FAME}$, (%) obtained with B-MOF-5 and T-MOF-5 catalysts after the contact with methanol.

A reaction mechanism was hypothesized, reported in Figure 8, in which the contact of the MOF-5 catalyst and methanol leads to a break of the bond between zinc and oxygen, and, at the same time, a bond with methanol itself is formed [41]. The hypothesis of the occurred mechanism is also confirmed by the results of the SEM analysis (Figure 3), in which the difference of the morphology before and after the contact with methanol can be seen. The images show the expected morphology of MOF-5 that is lost after contact with methanol, confirming the results of catalytic activities. The catalyst structure may cause diffusion problems due to the microporosity of the MOF-5. Operating with the catalyst in its open form can be advantageous in that it can avoid problems connected with the diffusion of the large bulk molecules as the triglycerides inside the catalyst itself. In Figure 8, a schematic representation, including part of the catalyst reported in Figure 6, is shown, and it can be extended to the whole structure.

The proposed mechanism for the MOF in the reactions studied retraces the one proposed by Sousa [41] and by Reinoso [42], who consider the nucleophilicity and the steric hindrance for the coordination at the Lewis site of zinc. In fact, considering the large size of the triglycerides, it is hypothesized that the coordination at the zinc site takes place by the action of methanol. After the methanol has broken the bond of the MOF, freeing the site, as evidenced by the XRD and results discussed above, this coordinates with the zinc site, with the formation of the Zn–OCH3 bond. Then, the carboxylate shift occurs with the formation of a carboxylic acid coordinate with $Zn^{2+}$. These groups, as also reported in the existing literature, correspond to Brønsted sites. Then, the polarization of the bond C=O of the triglyceride molecule absorbed by the methoxide takes place, and the last step is the formation of the methyl ester and diglyceride. For the esterification reaction, a similar mechanism is assumed to occur, where only methanol coordinates with zinc, whereas oleic acid and fatty acids are spectators.

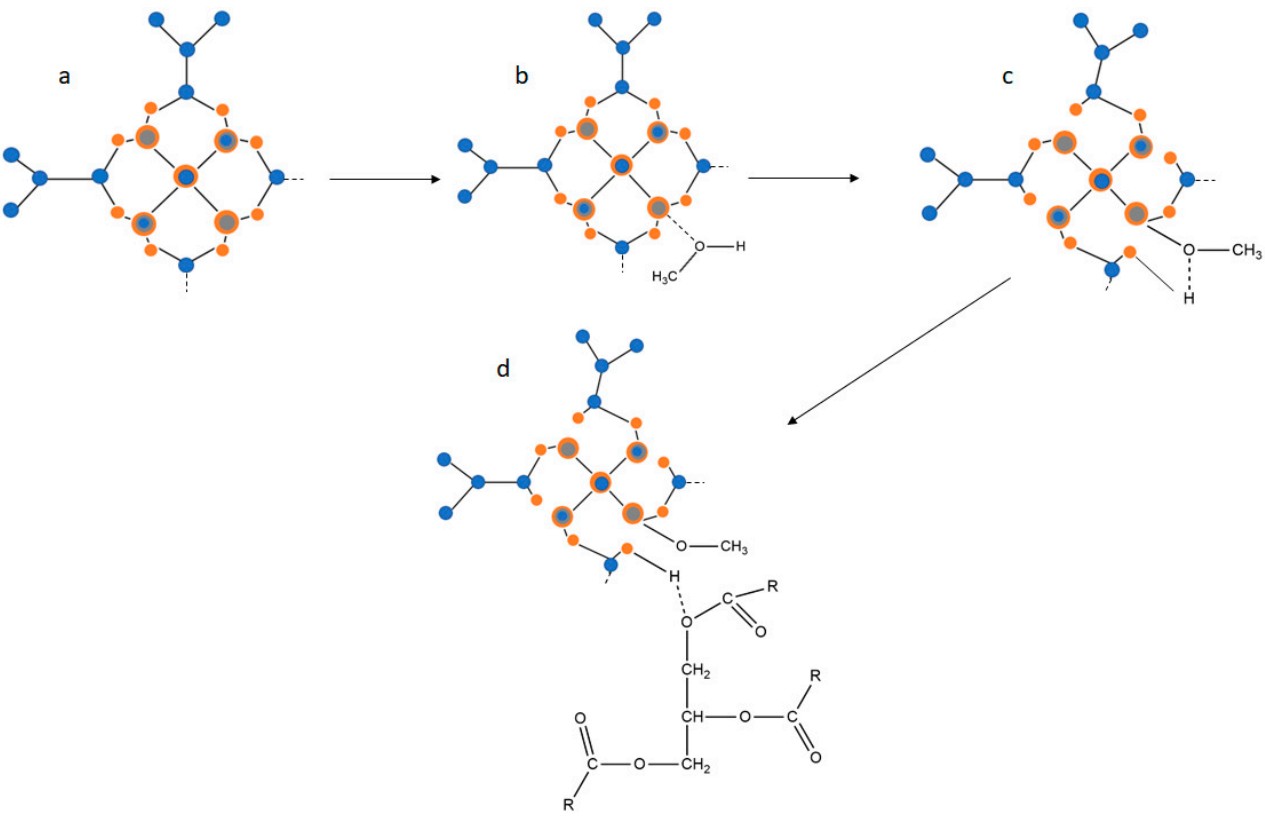

**Figure 8.** Reaction with MOF-5 in the presence of methanol: (**a**) MOF-5; (**b**); methanol coordination with ZnOOCH3 bond formation; (**c**) breakdown of the zinc-oxygen bond; (**d**) formation of active species. The atoms are represented in the following way: zinc in grey, oxygen in orange, carbon in blue.

## 3. Materials and Methods

### 3.1. Materials

The oil used in the runs was purchased in a local food store. The fatty acid composition, determined by gascromatographic analysis, was (wt%): palmitic = 11, oleic = 23, linoleic = 56, linolenic = 5, others = 1. All other employed reagents were supplied by Aldrich at the highest level of purity and used as received without further purification.

### 3.2. Synthesis of the MOF- 5 Based Solids

#### 3.2.1. Blend MOF-5

MOF-5 was prepared according to the method reported in [43]. Following this procedure, terephthalic acid ($H_2BCD$) and zinc nitrate hexahydrate ($Zn(NO_3)_2 \cdot 6H_2O$) were used as precursors for organic ligand and metal, respectively, whereas dimethylformamide (DMF) was applied as an organic solvent to build the skeletal of MOF. Particularly, a solid mixture containing $H_2BCD$ and $Zn(NO_3)_2$ (weight ratio 0.2) was dissolved in the solvent at room temperature. Then, the solution was heated until 130 °C and it was kept at this constant temperature for 4 h. During this crystallization step, the solution slowly changed from limpid to turbid due to the formation of a white solid. At last, after the filtration step, the crystals obtained were washed with acetone. The final step was the activation at 60 °C under vacuum overnight. The experimental conditions adopted for the synthesis are resumed in Table 2.

#### 3.2.2. Tetragonal MOF-5

The procedure reported by Zhang and Hu [37] and developed by Yaghi [44] was followed to synthesize MOF-5 with a tetragonal structure. This procedure consisted in

the direct mixing of reagents for the formation of the crystals. In particular, weighted amounts of $Zn(NO_3)_2$ and $H_2BDC$ (see Table 2) were dissolved in DMF as solvent at room temperature. Then, 10 drops of an aqueous solution of hydrogen peroxide (20 wt%) were added, and then 2.5 mL of triethylamine (TEA) was added using a syringe very slowly for about an hour. During this step, the solution slowly changed from limpid to turbid due to the formation of a white solid. Finally, the mixture was vacuum filtered in a nitrogen atmosphere and washed three times with DMF. The filtrate was then activated at 180 °C under vacuum for 24 h.

### 3.2.3. Cubic MOF-5 Synthesis

The cubic MOF-5 was synthesized following the procedure reported by Zhang [37]: 0.0829 g of $H_2BDC$ and 0.4593 g of $Zn(NO_3)_2$ were dissolved under vigorous stirring using a solution consisting of 49 mL of DMF and 1 mL of distilled water, at room temperature. After the complete dissolution of the reagents, the reaction mixture was heated to 100 °C and kept at this temperature value for 7 h. On heating, crystal formation was observed and after 7 h, the system was allowed to cool to room temperature. The solid was filtered and washed 6 times with 50 mL of DMF, each time leaving the solid in the DMF for 8 h. After the washing with DMF, the solvent was decanted and the solid was washed 6 times with 50 mL of chloroform, always leaving the solid in chloroform for 8 h. Finally, the chloroform was removed by decantation and the obtained solid was treated at 115 °C for 24 h under vacuum to activate it.

### 3.3. Characterization of the Catalysts

The X-ray diffraction pattern was determined with a powder diffractometer (Philips Analy-tical, 1887, Amsterdam, Netherlands), using CuKa radiation at 4 0 kV and 25 mA in a range 2°–70° (2-Theta) and at a scan rate of 0.01° (2 theta/s). Ultraviolet and visible light diffuse reflection (UV-DRs) spectra were recorded in the 200–800 nm on a double beam spectrophotometer (JASCO, V-550, Tokyo, Japan). Barium sulfate was used as a reflectance standard. The measured intensity was expressed as Kubelka–Munk function F(R). TGA analysis was performed with (Perkin Elmer, ST 6000, Columbus, OH, USA) heating the sample from 25 to 900 °C at 10 °C $min^{-1}$ in $N_2$ stream (20 $cm^3$ $min^{-1}$). The microstructures of the samples were investigated by scanning electron microscopy (SEM, NOVA/NANOSEM 450 FEI, Hillsboro, OR, USA).

### 3.4. Catalytic Runs

The simultaneous esterification and transesterification reactions were performed using steel vials reactors. The reactants, soybean oil or acid soybean oil (FFA = 10 wt%), and methanol, at a molar ratio of 3:1 and a defined amount of catalyst (5 wt%), were introduced in each reactor. The reaction conditions chosen are those that the research group has usually adopted to effectively study the activity of different catalysts for these reactions [11,14]. The reactors were introduced in a ventilated oven, in which the initial temperature was 50 °C and then increased at a rate 20 °C/min until the final reaction temperature (150 °C). Then, this temperature value was kept for three hours. At the end of the reaction, the reactors were quenched in a cold bath, and the mixture reaction was centrifuged at 3200 rpm for 20 min and then the solid catalyst was filtered. Finally, the excess of methanol was removed through evaporation. Blank runs (without catalyst) were also performed to evaluate any contribution of the non-catalytic reaction. The FAME yields were determined by NMR spectroscopy (Bruker 400 MHz, USA), measuring the area of the $^1$H-NMR signal related to methoxylic and methylenic groups [45]. The FFA conversions were determined by titration with a 0.1 M solution of KOH in ethanol [46].

### 3.5. Leaching Test

The leaching of Zn in the reaction system was evaluated through proper experiments. These runs consisted in two steps: at first, the catalyst was kept in contact with methanol,

without oil, using the same conditions used in the esterification/transesterification reactions for three hours. Then, the catalyst was removed by filtration and a new catalytic run was carried out with oil and without the catalyst, using methanol recovered from the previous run.

### 3.6. Reusability of the Catalyst

The reuse tests were performed to verify the stability of the solid catalyst. Each run was carried out in a Parr Instrument steel reactor using 20.5 g of acid oil and a molar ratio methanol/oil 80:1 and 0.125 g of MOF-5 (0.5 wt% respect to the oil), at 150 °C and 14 Bar as pressure for 3 h. After the reaction, the catalyst was separated from the reaction mixture by the decantation and immediately used in the consecutive reuse run by adding fresh oil and methanol to the reaction vessel.

### 3.7. Study of the Effect of Methanol on MOF-5 Structure

These runs were performed to verify the effective structure of the MOF in the transesterification reaction following contact with methanol and, therefore, to check the stability. For this purpose, 1 g of three MOFs (blend, cubic, and tetragonal) were put in contact with 20 g of methanol (99.8% for GC) in the steel vial reactors at three different temperatures (25 °C, 65 °C, and 150 °C) under stirring for 3 h. Finally, the solids recovered by filtration were dried in a vacuum oven at 40 °C. XRD and DR-UV analysis were performed on the dry samples.

## 4. Conclusions

In this work, the fundamental role of methanol in activating MOF-5 for simultaneous esterification and transesterification reactions to produce biodiesel from waste oil was highlighted. The methanol, used as a reactant for the reaction, was found to act to further open the structure of MOF-5 through structural distortion of the lattice. In each case, the active sites were created after contact with methanol by the breaking of the Zn–O bonds, increasing the accessibility of the triglyceride for the reaction. Therefore, this work sheds light on the mechanism by which MOF-5 catalyzes transesterification reactions, and paves the way for an improvement on the synthesis in order to make it more stable and active.

**Supplementary Materials:** The following supporting information can be downloaded at: https://www.mdpi.com/article/10.3390/catal13030503/s1, Figure S1: Catalyst reuse tests.

**Author Contributions:** Conceptualization, R.T. (Rosa Turco); methodology, R.V.; formal analysis, V.R.; investigation, R.T. (Rosa Turco); data curation, R.T. (Riccardo Tesser); writing—original draft preparation, R.T. (Rosa Turco); writing—review and editing, F.T. and M.D.S.; visualization, V.R. and R.T. (Riccardo Tesser); supervision, R.T. (Rosa Turco); project administration, M.D.S.; funding acquisition, M.D.S. All authors have read and agreed to the published version of the manuscript.

**Funding:** This research received no external funding.

**Data Availability Statement:** The data presented in this study are available on request from the corresponding author.

**Acknowledgments:** The authors acknowledge the grant PRIN: Progetti di Ricerca di Interesse Nazionale—Bando 2017—prot. number: 2017KBTK93 "CARDoon valorisation by InteGrAted biorefiNery (CARDIGAN)" of Italian Ministero dell'Istruzione dell'Università e della Ricerca for financial support. The authors also thank Valeria Vassallo for her valid support in experimental runs of the laboratory. Oreste Tarallo is acknowledged for the support in performing SEM analyses.

**Conflicts of Interest:** The authors declare no conflict of interest.

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
