# Peer review of "Biodiesel Production from Waste Oil Catalysed by Metal-Organic Framework (MOF-5): Insights on Activity and Mechanism"

_catalysts, doi:10.3390/catal13030503_

Round 1

Reviewer 1 Report

This manuscript includes some good catalytic performance of regular MOF-5 in the simultaneous transesterification and esterification of acid vegetable oils, and this work is interesting. However, there are several critical points that the authors need to answer before the manuscript could be considered for publication. Thus, I recommend a thorough major revision followed by further evaluation.

Authors are requested to revise the manuscript based on the following suggestions.

1. In the introduction, the novelty of the manuscript should be exhibited in comparison with other catalyst carriers (eg. metal oxide, hydrotalcites, porous polymeric, heteropolyacids, and MOF-derived materials) on synthesis of solid acid catalysts.

2. MOF-5 catalyst is very conventional and not novel, and there are many related research reports.

3. The structural characterization of the catalyst is too few, and the structure-activity relationship of catalysts is not discussed in detail.

4. The acidity and basicity of three different types of MOF-5 should be determined using some probe molecule.

5. The morphological images of three different types of MOF-5 should be added.

6. The author has to study the efficient in the esterification reaction for all conditions such as, percentage of oil to methanol, weight of catalysts, ……

7. The esterification and transesterification reactions have not been studied in kinetics (LHHW or ER) and thermodynamics.

8. All figures quality should be extensively improved.

Reviewer 2 Report

The authors reported the application of different types of a Zn-based MOF, namely MOF-5, for biodiesel production by esterification and trans-esterification reaction of acid vegetable oils. Such work is interesting, however, there are key points should be addressed before acceptance.

1)     One of the important notes lacking in the present work is a comparison table of the method (e.g. activity, selectivity, yield, porosity and….) with the other reports.

2)     I would have expected a stronger focus on mechanistic insights to the field, e.g. adding more evidence and discussing rather than observed here.

3)     The authors should provide a relevant figure for their esterification and trans-esterification reaction.

4)     The authors should discuss more and compare the different MOF design strategies, rather than just listing them.

5)     The authors should add IR, TGA, SEM/EDX, and PXRD analysis of all MOFs, discussing and comparing the differences. In addition, authors should merger the figures together for comparison.

6)     If possible, authors should provide the BET and porosity data for more evidence.

7)     I can’t find the reusability tests (yields and conversions) as mentioned in the text. Please add it to the text.

8)     The authors missed some of the recent studies of catalytic applications of porous MOFs in the literature (e.g., Chem. Soc. Rev. 2022, 51, 7810-7882; ACS Appl. Mater. Interfaces 2022, 14, 32, 36515-36526). I think they will be useful for more audiences and researchers working on porous materials.

9)     The language should be more accurate (e.g., Lines 47, 79, 84, 173, 189, 190).

Reviewer 3 Report

1. The characterization is not enough to confirm the successful preparation of samples; please give the SEM, which will be needed for the reader. 2. Please also provide the BET for the original MOF-5. 3. The three types of PXRD should be interrogated with the simulated one.

4. An important property of MOFs is their modularity: the pore dimension, shape, and chemical environment can be tailored by the choice of a couple of building blocks (metal and organic linker), in addition to the solvent, and how they are interconnected. This part should be updated some refs, such as Micropor. Mesopor. Mat, 341(2022) 112098; Inorg. Chem., 2017, 56, 10215−10219 and J. Mater. Chem. A, 2016, 4, 11630-11634.

5. The conclusion should be simplified and give more value information.

6. “In figure 6, a schematic representation including part of the catalyst reported in Figure 4 was showed, and it can be extended to the whole structure.” I found not any supporting data. Moreover, the fig6 is not convincingness.

7. “In this work, three solids based on MOF-5, with different structures were synthesized,” is it new structure?

Round 2

Reviewer 1 Report

The authors revised the manuscript carefully and improved the quality of the manuscript. I agree to accept the manuscript in its current form.

Reviewer 2 Report

The manuscript has improved for publication in Catalysts.

Reviewer 3 Report

accepted